# Effect of Dihydroquercetin During Long-Last Growth of *Yarrowia lipolytica* Yeast: Anti-Aging Potential and Hormetic Properties

**DOI:** 10.3390/ijms252312574

**Published:** 2024-11-22

**Authors:** Maxim S. Pusev, Olga I. Klein, Natalya N. Gessler, Galina P. Bachurina, Svetlana Yu. Filippovich, Elena P. Isakova, Yulia I. Deryabina

**Affiliations:** A.N. Bach Institute of Biochemistry, Research Center of Biotechnology of the Russian Academy of Sciences, Leninsky Ave. 33/2, Moscow 119071, Russiaklein_olga@list.ru (O.I.K.); gessler51@mail.ru (N.N.G.); galainbi@yandex.ru (G.P.B.); syf@inbi.ras.ru (S.Y.F.); yul_der@mail.ru (Y.I.D.)

**Keywords:** yeast, *Yarrowia lipolytica*, reactive oxygen species, polyphenols, dihydroquercetin, anti-aging potential

## Abstract

Polyphenols are powerful natural antioxidants with numerous biological activities. They change cell membrane permeability, interact with receptors, intracellular enzymes, and cell membrane transporters, and quench reactive oxygen species (ROS). *Yarrowia lipolytica* yeast, being similar to mammalian cells, can be used as a model to study their survival ability upon long-lasting cultivation, assaying the effect of dihydroquercetin polyphenol (DHQ). The complex assessment of the physiological features of the population assaying cell respiration, survival, ROS detection, and flow cytometry was used. *Y. lipolytica* showed signs of chronological aging by eight weeks of growth, namely a decrease in the cell number, and size, increased ROS generation, a decrease in colony-forming unit (CFU) and metabolic activity, and decreased respiratory rate and membrane potential. An amount of 150 µM DHQ decreased ROS generation at the 6-week growth stage upon adding an oxidant of 2,2′-azobis (2-amidinopropane) dihydrochloride (AAPH). Moreover, it decreased CFU at 1–4 weeks of cultivation, inhibited cell metabolic activity of the 24-h-old culture and stimulated that on 14–56 days of growth, induced the cell respiration rate in the 24-h-old culture, and blocked alternative mitochondrial oxidase at growth late stages. DHQ serves as a mild pro-oxidant on the first day of age-stimulating anti-stress protection. In the deep stationary stage, it can act as a powerful antioxidant, stabilizing cell redox status and reducing free radical oxidation in mitochondria. It provides a stable state of population. The hormetic effects of DHQ using lower eukaryotes of *Y. lipolytica* have been previously discussed, which can be used as a model organism for screening geroprotective compounds of natural origin.

## 1. Introduction

Many researchers are focused on the polyextremophilic yeast of *Yarrowia lipolytica*. The yeast possesses a unique ability to assimilate various substrates and maintain viability under unfavorable conditions, namely extreme pH, thermal shock, and the impact of toxic compounds [1]. The *Y. lipolytica* culture’s ability to prosper under extreme conditions is directly related to its ability to scavenge the reactive oxygen species (ROS). However, the aging process is tightly associated with ROS formation, the accumulation of damaged molecules, and mitochondria activity [2]. Mitochondria, especially in the functioning of the electron transport chain, where abundant generation of the superoxide anion radical O_2_^•−^ is undergone, are the main source of the ROS in the cell. Moreover, external factors such as hypoxia aggravate the process [3]. Some other natural processes producing the ROS, namely hydrogen peroxide (H_2_O_2_) and O_2_^•−^, include NADPH oxidases (NOX), unbound NO synthase, cytochrome P450, xanthine oxidase, endoplasmic reticulum, peroxidases, and cyclooxygenases [2]. Little concentrations of intracellular ROS are vital since they are involved in the fundamental processes of carboxylation, hydroxylation, peroxidation, and modulation of signal transmission pathways [4]. ROS have been described to mediate several intracellular pathways, including proliferation, inflammation, metabolism, and cell death. Recently described redox-sensitive kinases and transcription factors that mediate redox signaling reactions include Nrf-2, p38MAPK, HIF-1α, PGC-1, FOXO, and Nf-kB [5,6]. Excessive and persistent ROS cause oxidative stress and are associated with some pathologies. On the other hand, excessive reductive stress can also be fatal to the cell being associated with cancer, cardiomyopathy, and aging [7,8]. At the cellular and tissue levels, free radicals are involved in protection against pathogenic organisms, fertilization, cell fission, apoptosis, regeneration, cell movement, and regulation of vascular tone [4].

In its turn, mitochondria are the key participants in the aging of cells. As mentioned above, mitochondria are the main source of ROS in cells. However, mitochondria are a suitable target for free radicals, which affect mitochondrial DNA, resulting in the distorting of mitochondria protein biosynthesis, mitochondria dysfunction, and an increase in the ROS level, leading to cell aging [9]. A decrease in antioxidant protection accelerates aging. However, moderate ROS production induces “mitohormesis”, an adaptive response to moderate stress, which provides resistance to stress and extending the cell’s lifespan [10].

It is believed that the study of the molecular mechanisms of aging could reveal the reasons for complicated changes in tissues and organs. Thus, gerontological studies use unicellular models of eukaryotic cells, including yeast. Gerontological studies have been used to study many genetic and chemical factors that change the life expectancy of mammals [11,12]. Among yeast cultures, gerontological studies are performed mainly using baker’s yeast, *Saccharomyces cerevisiae*, due to the diversity of both collections with deleted genes and libraries with super-producer cells. The fast growth rate of the cultures and the ability to assimilate fairly simple nutrients and substrates give extra advantages [13]. The strains of harmless polyextremophilic yeast of *Y. lipolytica* are a popular producer in the industry. It has some features that distinguish it from *S. cerevisiae*, namely a complete respiratory chain containing all the same respiratory complexes as in mammals and plants. There are complexes I–IV, one “alternative” NADH-dehydrogenase (NDH_2_), and non-heme alternative terminal oxidase (AOX), which are activated at stress and in the stationary growth phase [14,15]. Since complexes I and III are responsible for the main contribution to generating the ROS upon electron transfer [16], the induction of the alternative oxidase decreases the active oxygen radical production [17].

*Y. lipolytica*’s ability to prosper under adverse conditions, namely under extreme pH (from 2.5 to 9.5), under increased salt concentration and toxic compounds, and under thermal shock (up to 38 °C) is well-known [18]. These features, unique even to most yeast, permit the use of the *Y. lipolytica* yeast as a model for protein secretion, assimilation of various substrates, processes of sexual dimorphism, and aging [14]. Therefore, *Y. lipolytica* can serve as an excellent tool for studying physiological responses to aging-induced oxidative stress and different antioxidant compounds. Natural polyphenols are the most widely used antioxidants in pharmaceuticals and para-pharmaceuticals [19,20]. They possess moderate hydrophobicity, which allows them to penetrate cell membranes and some other barriers under specific conditions. Some polyphenols carry charged groups and can change their hydrophobicity depending on pH. Polyphenols actively interact with lipids in different cell membranes, including plasma and mitochondria, with the former changing the lateral fluidity and permeability of the latter. Along with antioxidant activity, some polyphenols can specifically interact with some receptors on the cell surface, various cell enzymes (including those from the mitochondria respiratory complexes), and membrane ion channels (transporters) [21]. The antioxidant properties of flavonoids play an immense role in human health [22]. Various mechanisms of polyphenol action include both direct ROS quenching and chelation of metals with variable valence, as well as indirect effects of antioxidant system induction in the cell [23]. The hydroxyl groups of flavonoids and the ability to delocalize unpaired electrons forming several resonant structures permit the direct scavenging of ROS [24]. However, polyphenols can affect the animal cells damaged under adverse conditions as hormetic stressors. They can induce the cell protection systems aimed at controlling the redox state, proteostatic and metabolic homeostasis, inflammatory response, and some others, making cells more resistant to toxic stimuli, especially in the elderly [25]. However, the mechanisms of action of polyphenols are not known completely [26].

In this regard, one of the attractive objects for researchers is the representative of dihydroflavonols taxifolin (5,7,3′,4′-flavan-on-ol) or dihydroquercetin (DHQ). DHQ is a flavonol with two stereocenters on the C-ring and methylation at C-3, C-5, and C-7. It was first isolated from Douglas fir, *Pseudotsuga taxifolia*, and named after it. DHQ is widely distributed in medicinal plants such as *Allium cepa* L., *Silybum marianum* L., *Catha edulis*, and *Larix gmelinii*. The strong biological activity of DHQ, in particular its antibacterial, antiviral, and antitumor properties, determines its pharmaceutical value [27]. The protective effect of DHQ on the redox status of the cell is associated with its antioxidant properties, which are not inferior to the well-known antioxidant α-tocopherol and are manifested in the inhibition of the production of superoxide anion radical, blocking lipid peroxidation in mitochondria and activation of NADPH-dependent cytochrome P450 [28]. However, despite its high scientific and practical interest, the precise mechanisms of DHQ action on cellular components have not yet been studied. In this regard, it is of particular interest to study the ability to maintain viability during prolonged cultivation of an aging culture and to evaluate the effect of DHQ under these conditions using a model of unicellular eukaryotes, yeast *Y. lipolytica*, which has a high similarity to mammalian cells. The contribution of our research in this direction allows us to better understand the mechanisms of the antioxidant action of polyphenols in the development of cellular pathologies associated with aging.

## 2. Results

### 2.1. The Dynamics of the Cell Number and Their Morphology

Counting the total number of cells and assaying the share of budding cells showed that the population reached the highest number of cells, (50 ± 16) × 10^10^, by 24 h of cultivation. In the first week of growth, their number decreased to (10 ± 5) × 10^10^, whereas from the second week and up to the eighth one, the average number of the cells remained constant at the level of (6 ± 2) × 10^10^ cells (Figure 1a). However, the number of budding cells reached 20–25% of the total one on the first day of cultivation, increased to 35–40% on days 7 and 14, followed by a decrease to nearly 30% and even 20% in 4 and 6 weeks, respectively (Figure 1b). It is of interest that approximately 30± 10% of budding cells of the total one remained upon complete nutrient depletion in 8 weeks of growth (Figure 1b). Just from the first day of cultivation, the cells were round and had some vacuoles (Figure 1c). At the same time, during prolonged cultivation, the degree of cytosol vacuolization increased significantly (Figure 1d). There was no significant effect of DHQ on the total number of cells in the population and the frequency of cell budding, with the exception of the 7-day and 14-day growth stages, where the introduction of DHQ caused a multiple decrease in the number of budding cells by 8- and 2.4-fold, respectively (Figure 1b).

The flow cytometry method (Figure 2) showed a decrease in the forward scatter (FSC) signal, which reflected a decrease in the size of *Y. lipolytica* cells upon long-lasting growth (Figure 2a) [29]. The data analysis showed that by 8 weeks of cultivation, cell size gradually decreased to 72% compared to those of a 24-h culture. The side scatter (SSC) signal correlates with the cytosol granularity of the cells [29]. It decreased by 10–15% in the first two weeks of growth, followed by an increase up to 134% by eight weeks of cultivation compared to that in the 24-h population (Figure 2b). According to the results in Figure 2 (diagrams), beginning from the first week of growth, the *Y. lipolytica* culture formed two subpopulations: with low (the left shoulder) and high (the right shoulder) light scattering features that corresponded to relatively small and large cell sizes. By the fourth week, the share of small cells had increased, but it significantly decreased by the sixth week, which, presumably, could indicate the death of some cells in the population of small ones accompanied by the use of the nutrients from the dead cells to nourish new cells.

There was no evident effect of DHQ on the total number of the cells in the population, the share of budding cells (Figure 1), the size (forward scatter signal) and granularity (side scatter) of the cells (Figure 2).

### 2.2. Dynamics of the Reactive Oxygen Species Level

The level of the cellular ROS was assayed using a non-fluorescent derivative of fluorescein 2′,7′-dichlorodihydrofluorescein diacetate (H_2_DCFDA). As the positive control for the ROS generation, hydrophilic AAPH, a hydrophobic oxidizer, was used, which upon entering the cell spontaneously forms the conjugated diene hydroperoxides and induces the lipid oxidation [30]. The fluorescence on the seventh day of cultivation decreased to 82 ± 10% compared to the one-day-old culture of *Y. lipolytica*, which increased at 28 days of cultivation and reached 173 ± 27% on the 42-th day. Then, the ROS level decreased to 74 ± 31% by the eighth week. The action of AAPH showed a similar trend (the increased fluorescence signal by the 42nd day of cultivation, followed by the decrease) (Figure 3a).

The assay of the DHQ showed that the polyphenol affected no intracellular ROS when administered only with H_2_DCFDA, except for at the 6-weeks stage, when the ROS generation decreased by 24% (Figure 3). However, there was a tendency to decrease the ROS produced by the oxidant AAPH upon the DHQ action compared to that of the control samples in the *Y. lipolytica* cells of the same age within the same experiment (Figure 3b). In 2 and 4 weeks of growth, the ROS generation decreased by 8%, while in a 6-week-old population, DHQ halved the oxygen radical production (Figure 3b).

### 2.3. Dynamics of the Number of CFU and Metabolic Activity

Upon the long-lasting cultivation, the number of CFU to the total cell number in the *Y. lipolytica* population decreased from 63 ± 15% (one week of cultivation) to 21 ± 3% by 8 weeks (Figure 4). However, the greatest decrease in the CFU number (by 36%) was noted in the fourth week of cultivation compared to that at the second week of growth. DHQ reduced the number of CFUs at 1–4 weeks of cultivation, on average, by 20%; however, in the 6-week-old and 8-week-old populations, the number of CFUs in the experimental samples coincided with that in the control ones (Figure 4).

The dynamics of the metabolic activity (viability) of the *Y. lipolytica* yeast upon long-lasting cultivation were assayed using an 3-(4,5-dimethylthiazol-2-yl)-2,5-diphenyltetrazolium bromide (MTT) test (Figure 5). The viability assessment showed a sharp decrease in metabolic activity by the seventh day of cultivation by 60 ± 11%, reaching 72 ± 9% by the eighth week of growth (Figure 5a). Assaying the optical density (Figure 5a) showed the coincidence of the metabolic activity with DHQ addition with that in control throughout the experiment, except for the 14-day-old population, where metabolic activity in the samples treated with DHQ doubled. Moreover, if we consider the effect of DHQ as the ratio of the optical density of the experimental samples to the control zones of the same age, DHQ inhibited the one-day-old culture by 25% and stimulated it by 1.9 times on the 14-day-old population. As for the older culture, its metabolic activity was induced by 35%, 25%, and 15% by the 28, 42, and 56 days of growth, respectively (Figure 5b).

### 2.4. Changes in the Mitochondria Activity

#### 2.4.1. Respiratory Activity of *Y. lipolytica* Cells

The respiratory activity was assayed with the polarographic method. To induce respiration, a cell suspension was done in a 50 mM KP_i_ buffer with 25 mM glucose as a substrate. In the mitochondria of numerous organisms including all plants, most fungi, algae, and some protozoa tested so far, besides the canonical cyanide-sensitive cytochrome oxidase, the respiratory chain contains a cyanide-insensitive, hydroxamic acid-sensitive terminal oxidase called AOX [31]. Electronic transport through AOX is not coupled with ATP synthesis and energy conservation; thus, the energy of ubiquinol oxidation by oxygen is released as heat. The cyanide-sensitive respiration through the main cytochrome pathway was inhibited by 4 mM KCN application, and the alternative cyanide-insensitive respiration was inhibited by 4 mM salicylhydroxamic acid (SHAM) (Figure 6).

The respiratory rate decreased by nearly 43 ± 15% on the seventh day of cultivation, which correlated well with a decrease in metabolic activity (Figure 5). On the 56th day of the cultivation, the respiration rate declined to 28 ± 1% compared to that in the 1-day-old yeast. Cyanide resistance of the respiration was assayed in the 1- and 8-week-old cultures, which indicated the induction of alternative respiration. However, the 1-week-old culture showed the highest resistance to KCN and SHAM, and the respiratory inhibition did not exceed 48%. Subsequently, the combined action of two inhibitors facilitated inhibition up to the complete block of respiration by the eighth week of cultivation (Table 1).

We showed that DHQ increased the respiration rate in daily culture, up to 40% compared to control samples, without significantly affecting the respiratory activity of cells at later growth stages (Figure 6). However, an assay of the alternative oxidase activity showed that the polyphenol decreased by about 10% on the first day of growth. On the other hand, its induction decreased by 2.8, 1.9, and 2.4 times in the 1-, 2- and 8-week-old cultures compared to that in the control samples, respectively (Table 1).

#### 2.4.2. Potentiometric Staining of the *Y. lipolytica* Cells with Analysis Using Flow Cytometry

The mitochondrial potential was assessed with flow cytometry. The cells were stained in vivo with potentiometric JC-1, a combination of MitoTracker Green and MitoTracker Red molecular probes. The JC-1 dye initially exists in monomeric form (wavelength of green fluorescence of 529 nm) and selectively accumulates in mitochondria and the accumulation increases with rising the mitochondrial potential [32]. High dye concentration (more than 0.1 µM in aqueous solutions) and a high mitochondrial potential induced forming J aggregates from the molecules and the emission wavelength changes to red (590 nm). Thus, the mitochondrial potential is assayed based on the ratio of the red fluorescence intensity to green fluorescence [33].

The ratio of the red fluorescence of JC-1 to the green one increased on the seventh and fourteenth days of cultivation (Figure 7). Beginning from the 42nd day, mitochondrial potential decreased to 33 ± 8% compared to that in the 1-day-old population due to both an increase in the intensity of the green fluorescence signal and the increased number of the cells with a low intensity of the red one. Staining with JC-1 showed that DHQ increased the mitochondrial potential by 46 ± 7% on the first day of cultivation, but later did not affect it (Figure 7).

The MitoTracker Green FM molecular probe (excitation/emission = 490/516 nm) accumulates in the active mitochondria. In the solutions, the dye does not glow, but being bound to the mitochondria, it gives a green glow. Presumably, the dye mainly accumulates in the mitochondria matrix where it covalently binds to mitochondrial proteins, reacting with free thiol groups of cysteine residues regardless of the membrane potential that permits to assay the total mitochondria volume in the cell [33]. The MitoTracker Red FM molecular probe (excitation/emission = 581/644 nm) accumulating in the active mitochondria depends on the membrane potential and causes a bright red glow. Only the mitochondria generating membrane potential and maintaining membrane polarity are stained [34]. Figure 8, Figure 9 and Figure 10 show the results of the culture mitochondria state (10,000 events) upon staining with MitoTracker Green and Red dyes. According to the data obtained, the mitochondrial volume decreased up to the fourth week (up to 47 ± 15% compared to that in 1-day-old culture); however, by the eighth week, it reached the control level (24 h of cultivation) (Figure 8a,b). The cell mitochondrial potential decreased (67 ± 10% compared to that in the 1-day-old culture), whereas, at 6 and 8 weeks of cultivation, the culture potential reached a similar level on the first day of growth and even exceeded it (Figure 8a,b).

The results suggested that the increase in the mitochondrial activity by the eighth week was associated with some decrease in the number of cells with low mitochondrial volume (number of free thiol groups) and mitochondrial potential. The identification of the subpopulations with low mitochondrial activity (P2) and high mitochondrial activity (P1) (Figure 9a) permitted us to quantitively assay the alterations in the cell distribution upon long-lasting cultivation. It is noteworthy that the state of mitochondrial subpopulations also changed depending on the growth time (Figure 10).

The combination of MitoTracker Green and MitoTracker Red molecular probes in our experiments also showed a significant decrease in both mitochondrial volume and mitochondrial potential of the experimental samples compared to that in the daily control culture (Figure 8). The distribution of the subpopulations of the cells grown with DHQ addition was performed similarly to that in the control samples. The distribution of the cells in the subpopulations showed that up to the sixth week of the culture growth, the number of cells with a high potential and high mitochondrial activity (P1) in the control and experimental samples remained approximately the same. However, at 6 and 8 weeks of growth with DHQ addition, the number of cells with high mitochondrial activity increased by 26% and 23%, respectively (Figure 9b).

Moreover, the shapes of the peaks for both MitoTracker Green and MitoTracker Red signals coincided in the control and experimental samples, but the signal intensity differed (Figure 10).

## 3. Discussion

DHQ (1,3,5,7,3′,4′-pentahydroxyflavanone), also known as taxifolin, is a flavonoid with high biological activity, widely spread in the plants of the Pinaceae family, such as the Siberian larch, Douglas fir, and Himalayan cedar [35]. Also, DHQ milk thistle and onion seeds contain a lot of DHQ [36,37]. DHQ has been listed as a new food raw material at the international level and has been approved as a biological health supplement in food and medicine [38]. Moreover, DHQ has always been used for the treatment of cardiovascular and cerebrovascular diseases [39]. Numerous studies have shown the pharmacological activity of DHQ, including anti-cancer [40], anti-inflammatory [41], antioxidant [42], antiviral [43], anti-bacterial [44], anti-diabetic [45], and neuroprotective effects [38]. An ability to decline apoptosis upon naturally occurring neuronal death has been revealed [46]. Moreover, it can prevent the apoptosis of retinal pigment epithelium cells caused by oxidative stress [47]. Also, hepatoprotective [48] and anti-fungal activity of DHQ has been shown [49]. It is also used in various commercial drugs such as Legalon™, Pycnogenol^®^, and Venoruton^®^.

Thus, it is extremely promising to study the effect of DHQ, which is part of the human diet, on various pathologies associated with aging processes. Therefore, the use of yeast cell models seems most relevant. Some effects of polyphenol-containing plant extracts from various sources have been performed using yeast models. Thus, some screening tests of plant extracts have demonstrated their positive effects on chronological life expectancy and the mechanisms related to aging in the fission yeast of *S. cerevisiae*. The efficacy of fifteen extracts from *Serenoa repens* berries, aboveground parts of *Hypericum perforatum*, *Ilex paraguariensis* and *Ocimum tenuiflorum* leaves, *Humulus lupulus* and the whole plant of *Solidago virgaurea*, *Citrus sinensis* fruits, *Vitis vinifera* grape peel, *Andrographis paniculata* whole plant, *Hydrastis canadensis* roots, *Trigonella foenum-graecum* seeds, the leaves of *Berberis vulgaris*, the flowers and stems of *Crataegus monogyna*, leaves of *Taraxacum erythrospermum*, and the whole plant of *Ilex paraguariensis*, extended the chronological lifespan of the yeast [50]. The fifteen extracts have been shown to play a significant role in cell processes, namely enhanced mitochondrial respiration, decreased levels of cell ROS, protection of genetic material and proteins from oxidative damage, and increased resistance to oxidative and thermal stress. Highly conserved signaling pathways, which regulate life expectancy, namely the rapamycin 1 (TORC1) complex, are the targets of the plant extracts tested [51]. Another is the protein kinase A (PKA) pathway [52] and Pkb-activating kinase homolog (PKH1/2) pathway [53], which prolongs aging, as well as the sucrose non-fermenting (SNF1) pathway [54] and autophagy pathway [55], which also prolong aging [56]. In a recent paper by Kwong et al. [57], the authors showed the discovery of two new extracts from the leaves of *Manihot esculenta* and *Wodyetia bifurcata*, which extend the chronological lifespan of *S. cerevisiae* in a dose-dependent manner, which facilitates cells’ resistance to oxidative stress via inducing the stress response pathways. Besides the actions of plant extracts, the effect of special polyphenols on cells has been described, in particular, in some studies of yeast cell aging. In an early paper [58], the protective effects of pyrogallol, phloroglucin, and myricetin were tested on oxidative stress using *S. cerevisiae*. It was shown that myricetin extended the chronological lifespan of the yeast without mitochondrial superoxide dismutase (Sod2p), which exhibited a phenotype of premature aging and sensitivity to oxidative stress. It was also found that polyphenol of magnolol protected the *S. cerevisiae* yeast deficient in antioxidant protection genes from oxidative stress caused by prooxidants H_2_O_2_ and menadione. Magnolol was reported to reduce the ROS generation in the cell, decrease lipid oxidation, and increase glutathione, and it also extended the chronological aging of the culture [59]. Quercetin extended the chronological aging of yeast cells, providing trehalose and glycerol accumulation in the cells and decreasing the ethanol and acetate formation [60]. Also, the flavonoid neo-hesperidin was shown to prolong the chronological aging of yeast via decreasing ROS production, showing synergism with another citrus flavonoid, hesperetin [61]. On the other hand, in the paper [62], a decrease in the chronological lifespan (CLS) of the *S. cerevisiae* yeast with 100 µM of resveratrol addition was demonstrated, related to the prooxidant mechanism of its action.

The DHQ effect using a yeast model has not been sufficiently studied yet. The anti-fungal activity of DHQ was assayed using five different species of pathogenic fungi, namely *Alternaria alternata* (Fr.) Kessler, *Aspergillus fumigatus* Fresenius, *Aspergillus niger* van Tieghem, *Macrophomina phaseolina* (Tassi) Goid., and *Penicillium citrii*. The strains were treated with different concentrations (100, 300, 500, 700, 900, and 1000 ppm) of polyphenol, which showed high antifungal activity via significant inhibition of fungal growth depending on the dose [63].

In our study, we attempted to thoroughly assay the DHQ effects on some features of the physiology of the *Y. lipolytica* yeast, i.e., morphometry, ROS generation, colony forming units, metabolic activity, and mitochondrial functions, upon long-lasting cultivation. Throughout the experiment, we observed some signs of chronological aging of the culture, namely a nearly 10-fold decrease in the cell number by the eighth week of growth compared to that in the 24-h-old culture, a decrease in cell size and an increase in their cytosol granularity, an increased level of ROS, a decrease in CFU and metabolic activity, and a decreased respiration rate and membrane potential. All the signs matched the main criteria of chronological aging, which manifests the inhibition of yeast growth while reaching a critical state, usually due to the depletion of essential nutrients and the accumulation of toxic metabolites in the cultural medium [64]. We found some interesting effects of DHQ, displayed in different ways at different stages of the cultivation of the *Y. lipolytica* yeast.

DHQ decreased the ROS generation by 24% at the 6-week growth stage (Figure 3). However, polyphenol facilitated the decrease in ROS generation in 2 and 4 weeks of cultivation with the applied oxidant of AAPH compared to that in the control samples of the same age (Figure 3b). Thus, presumably, the antioxidant properties of DHQ are typical of flavonoids, as previously studied using AAPH in the work [30]. AAPH, being a hydrophilic azo-radical initiator, forms coupled diene hydroperoxides via spontaneous decomposition at 37 °C. The radicals generated immediately react with oxygen and cause lipid oxidation. The initial antioxidants, acting as water donors, bind free radicals and delay cell lipid oxidation. The protective effect of DHQ on the cell redox status is related to its powerful antioxidant properties, comparable to a well-known antioxidant of α-tocopherol. It can inhibit superoxide anion production, blocking lipid peroxidation in the mitochondria and inducing NADPH–dependent cytochrome P450 [27,65]. Probably, the peculiarities of the polyphenol chemical structure, which is of antioxidant potential, namely the o-dihydroxy structure in the B-ring, which provides molecule stability, and the 5- and 7-OH groups with 4-oxo functions in the A- and C-rings, responsible for the maximum “radical-trapping” potential, can make it unique [66]. Based on the data of a probable antiradical mechanism of DHQ action, we could suppose that AAPH, causing a significant hyperoxidation uncompensated by the antioxidant protection in the aging *Y. lipolytica* cells, induces some supplementary antiradical protection while it directly interacts with the cell lipids and prevents the set chain reactions of lipid peroxidation.

We also showed that DHQ decreased the number of CFU at the 1- to 4-week growth stage but did not affect it in the later stages of cultivation (Figure 4). In addition, DHQ caused a multiple decrease in the number of budding cells at the 7-day and 14-day growth stages (Figure 1b). However, the data on the effect of polyphenols on the survival of model yeast is extremely controversial. So, Ramos-Gomez, with his team [67], showed that the natural flavonoid of resveratrol shortened the yeast lifespan and provoked mitochondrial dysfunction in *S. cerevisiae* at a concentration of 100 µM. Similar data were obtained using baker’s yeast cultivated at substrate deficiency [68]. Also, 100 µM resveratrol caused an approximately 30% drop in culture survival upon limiting the glucose level to 0.1%. Perhaps, in the deep stationary growth stage (from the first to the fourth week of cultivation), upon acute depletion of substrates for cell growth and fission, DHQ may launch a population energy-saving mode when fewer new cells create favorable conditions for the whole population. Thus, the dose-dependent effect of resveratrol (25–200 µg per ml) upon 4-hour treatment led to the inhibition of the growth rate, cell fission, and full reconstruction of the *Schizosaccharomyces pombe* transcriptome and metabolome [69]. Moreover, in the group of genes with lower expression, there appeared “cell cycle genes” (with a decrease in expression by 57.14%), while in the group of genes with higher expression, there appeared “core genes of response to stress” (increase in expression by 40.54%). The inhibitory DHQ effect on the cell metabolic activity on the first day of cultivation and the stimulating effect on the 14th–56th days of growth could be explained similarly (Figure 5b). In our previous study [49], it was shown that 30 µM DHQ decreased the survival of the daily cells by about 25%. Thus, DHQ may have a mild prooxidant effect on the 1-day-old culture [70]), inducing the anti-stress mechanisms of cell defense; for instance, the activity of antioxidant enzymes (SOD and catalase) and alternative mitochondrial oxidase. At the cellular level, it resulted in a decline in the number of viable cells upon a stable state of the whole cell population at deep stationary growth stages. So, Moreira et al. [71] reported both the prooxidant and antioxidant effects of resveratrol using isolated brain and liver mitochondria from rats of both sexes. Nevertheless, some pro-oxidant effects of polyphenol were reported to trigger the stimulation of cell antioxidant protection via signaling pathways, which involved nuclear factors such as (erythroid-derived 2)-like 2 (ARE/Nrf2) and phosphatidylinositol 4,5-biphosphate 3-kinase/Akt (PI3K/Akt). This assumption also agrees with the fact that an increased mitochondria activity at the 6- and 8-week growth stages in the samples exposed to polyphenol was related to a decline in the cell number with decreased mitochondrial volume (the number of free thiol groups) and mitochondrial potential (Figure 9b).

A change in respiratory activity is the main feature of cell energy status. An alternative electron transfer pathway, which is induced by the inhibitors of the cytochrome pathway, namely KCN, azide, and antimycin A, is known to be located in the mitochondria of most plants, fungi, and yeast [72]. The switching of the electron flow is undergone at the level of reduced ubiquinone. It is specifically inhibited by benzohydroxamic acid production. It is generally believed that an alternative oxidase in the mitochondria of fungi and yeast is in inactive form under normal (non-stimulated) conditions. Its activity is induced by the action of mononucleotides (AMP, GMP). Since polyphenol components affect the cell membrane functions and the antioxidant status of cells, it is pivotal to study the respiratory activity and induction of the alternative yeast oxidase upon assaying their effects. We showed that DHQ activated the respiration rate in the 1-day-old culture by 40% compared to that in the control samples, but later it had no effect on the oxygen consumption (Figure 6). The effect was confirmed by potentiometric staining with JC-1 dye, which also demonstrated an increase in mitochondria potential by 46 ± 7% on the first day of growth (Figure 7). Perhaps the polyphenol could indirectly affect the components of the mitochondrial respiratory chain, gently promoting its activity and facilitating cell respiration. Thus, the isoflavonoid of hesperidin improves the enzymatic status of complexes I–IV using animal models [73]. Besides, in the paper [68], the authors showed a more than 2-fold increase in the respiration rate of the *S.cerevisiae* cells cultured with 5% glucose and 1 mM resveratrol. Probably, a similar mechanism may work in our experiments.

Also, an assay of the alternative oxidase activity showed that the polyphenol application decreased it by about 10% in the first days of growth of the *Y. lipolytica* yeast, whereas its induction decreased by 2.8, 1.9, and 2.4 times at the first, second, and eighth weeks of cultivation compared to that in the control, respectively (Table 1). It is known that, in different organisms, there is an extensive system of alternative oxidase regulation, likely via a common signal including the ROS generation (in particular, superoxide and hydrogen peroxide), the influence of some acids (citrate, acetate, salicylate), ethanol, cAMP, and Ca^2+^ ions [72]. It has also been established that the cyanide-resistant electron transfer pathway in yeast is an indicator of cell stress [31]. Considering all the arguments and the fact that the polyphenol effects manifest upon a jump in the ROS production (Figure 3a), we could suppose that DHQ in the concentration tested at the deep stationary stage acts as a powerful antioxidant stabilizing the cell redox state, decreasing the free-radical oxidation in the mitochondria and thereby inhibiting the alternative oxidase. This can be confirmed by our studies, where we showed that DHQ, along with the polyphenols resveratrol, pinosilvin, and dihydromyricetine, suppressed ROS production using the isolated mitochondria of the *Dipodascus (Endomyces) magnusii* fungus [74]. We demonstrated that the inhibition of ROS generation was associated with both a block of oxygen consumption and the antioxidant features of the polyphenols. Another essential aspect follows from the results obtained. Our experimental model suggested a polyphenol supplement once a week, beginning from the first day of cultivation. Therefore, the effects obtained played into the theory of hormesis, when trace concentrations of a compound favorably affect the yeast viability due to weak stimulating stress, whereas the increased concentrations decrease the cell viability due to exceeding the toxicity threshold [75]. The phenomenon was described in [55], where 35 plant polyphenolic extracts were screened, and 6 of them, including the extracts from the roots and rhizome of *Cimicifuga racemosa*, *Valeriana officinalis* L. root, and *Ginkgo biloba* leaves, increased the chronological lifespan of yeast via various ways concerning the longevity. There are some effects of the extracts containing gingerols, epigallocatechin gallate, puerarin, silymarin, lutein, and piperine, i.e., an increase in the mitochondria respiration and membrane potential, some change in the cell ROS generation, reduction of oxidized molecules, an increase in resistance to oxidative and thermal stress, and the increased the decay rate of neutral lipids in lipid drops. However, according to the authors, geroprotective effects were due to hormesis. The hormetic effects of chrysin and apigenin, which temporarily increase ROS production and trigger an adaptive response via mitohormesis, increase the oxidative stress provided cell metabolic adaptation, finally extending longevity in *Caenorhabditis elegans* [76]. It was shown that some low molecular weight compounds, namely epigallocatechin gallate and epicatechin gallate from green tea, extended life expectancy via mitohormesis [77]. The molecules of plant origin could be considered xeno-hermetic compounds, which can be beneficial for plants directly or inducing their stress protection pathways. Concerning mitochormesis, in vivo polyphenols are believed to indirectly affect intracellular oxidative stress via activating the major antioxidant enzymes, namely superoxide dismutase, catalase, and glutathione peroxidase [78]. We suppose that DHQ may have similar effects. Our previous data shows that at thermal stress, the addition of 300 µM DHQ to *Y. lipolytica* cells caused a nearly 1.5-fold induction of superoxide dismutase activity, and an increase in catalase activity by 46% confirmed the assumption [49]. The hypothesis is confirmed by a significant increase in the respiration rate in the 1-day-old cells grown with DHQ (Figure 6) that may reflect the slight uncoupling effect of polyphenol, as happens with polyphenols with chrysin and apigenin [79].

## 4. Materials and Methods

### 4.1. Yeast Strains and Growth Conditions

The wild-type yeast *Y. lipolytica* W 29 from the collection of CIRM Levures (France) was used in the study. The culture was raised in batches of 100 mL in medium containing (g/L) MgSO_4_—0.5, (NH_4_)_2_SO_4_—0.3, KH_2_PO_4_—2.0, K_2_HPO_4_—0.5, NaCl—0.1, and CaCl_2_—0.05, with glycerol—10.0 as a substrate. Then, 2 M KP_i_ stock buffer was prepared by dissolving KH_2_PO_4_ anhydrous (272 g/L, Amresco, Boise, Idaho, USA, Cat # 0781), pH = 6.0. Further, 2 M KP_i_ stock buffer was prepared by dissolving K_2_HPO_4_ anhydrous (342 g/L, Amresco, Boise, Idaho, USA, Cat # 0705), pH = 9.0. Both KP_i_ buffers were sterilized by autoclaving and added to the sterile medium (ratio 1:40) just before inoculation. The yeast was cultivated at an ambient pH of 5.5 and a temperature of 28 °C, as described in [80]. The yeast was cultivated from 24 h to 8 weeks using a laboratory shaker with constant stirring of the flasks (155 rpm) and maintaining a constant temperature of 28 °C. DHQ was provided by TransMIT GmbH of the PlantMetaChem Group (Giessen, Germany). A 100 mM DHQ solution was prepared by dissolving 30 mg in 1 mL of DMSO. Then, 150 µL of the solution was added to half of the flasks (150 µM of DHQ), and 150 µL of DMSO was added to the second half of the flasks as the control. Every 7 days, DMSO and DHQ were added to controls and experimental samples under aseptic conditions. Absorbance (A) was assessed in cell suspension at the wavelength of 590 nm (A_590_) using a Specol-11 spectrophotometer (Zeiss, Jena, Germany).

### 4.2. Cell Respiration

Oxygen consumption by yeast cells was assessed in vitro at 25 °C using oxygen Clarke electrodes coated with a fluoroplastic film at a constant potential of −660 mV. The incubation medium for the experiment contained 50 mM KPi at a pH of 5.5 and 1% glucose [81].

### 4.3. Detection of ROS

The dynamics of intracellular total ROS production were monitored using a spectroscopic fluorescence probe of dihydro-2, 7-dichlorofluorescein diacetate ester (H_2_DCF-DA) (Sigma, Saint-Luis, MO, USA), as described previously [81].

### 4.4. MTT-Test

The *MTT* assay is a colorimetric assay for assessing cell metabolic activity by reducing the tetrazolium dye MTT (Wuhan Fine Biotech, Hubei, China), to its insoluble formazan, which has a purple color. First, 0.4 mL of tetrazolium dye of MTT solution with a concentration of 1 mg/mL of PBS was added to a cell suspension with a concentration of 5 × 10^9^ cells/mL of PBS in the ratio 1:1. Incubation was performed in the dark under conditions similar to cultivation for an hour. The cell suspension was then centrifuged at 12,000× *g* for 5 min at 4 °C. The supernatant was removed, and the cells were resuspended in 1.2 mL of isopropanol and subjected to ultrasound treatment for 3 min (at an amplitude of 5 microns). The samples were centrifuged under the same conditions; then, 200 µL of the supernatant was transferred to the wells of the tablet and assayed at an absorption wavelength of 565 nm using a spectrophotometer.

### 4.5. Flow Cytometry

Cells were centrifuged and resuspended in a medium to determine cell concentration. Cells (1 mL) were transferred into a 15 mL conical tube, centrifuged at 3000× *g* for 10 min, and the suspension was diluted to an optical density (590 nm) of 2.0. Mitochondrial functionality was assessed by a cytometric method using intravital cell co-staining with MitoTracker™ Red and MitoTracker™ Green dyes. Yeast cells were washed in PBS solution, the suspension was diluted to an optical density (590 nm) of 2.0, and dyes were added in equal volumes to a final concentration of 100 nM. Samples were incubated in the dark for 30 min at room temperature; then, a dilution was prepared according to an optical density of 0.1 and analyzed on a Beckman CytoFLEX^®^ V2B2Y2R0 cytometer (Beckman Coulter, Brea, CA, USA). The data obtained were analyzed using the CytExpert software (version 2.4.0.28) and visualized using the FlowJo software (version 10.8.1).

### 4.6. Potential-Dependent Staining

Potential-dependent staining of mitochondria in the *Y. lipolytica* cells raised under different conditions was performed using molecular probes of JC-1, MitoTracker™ Red, and MitoTracker™ Green (Thermo Fisher Scientific, Waltem, MA, USA). Cells were incubated with 0.5 µM dyes and examined immediately and in 15, 20, and 30 min. The incubation medium contained 0.01 M PBS, pH 7.4, supplemented with 1% glycerol. Regions of high mitochondrial polarization are indicated by red fluorescence due to the concentrated dye. To examine the MitoTracker™ Red-stained preparations, filters 02 and 15 (Zeiss, Oberkochen, Germany) were used (magnification × 100). Photos were taken using an AxioCam MRC camera (Zeiss, Oberkochen, Germany).

### 4.7. Assay of the Protein Amount

Total protein was assayed by both the biuret method and the Bradford method with BSA as a standard. The optical density of solutions was assayed on a spectrophotometer at 595 nm.

### 4.8. Statistical Analysis

The experiments were performed in biological triplicates with a standard error of less than 5%, with two analytical replications. The results of the experiments were compared to the control. The figures show the arithmetic mean values of cell numbers and their standard errors. The method of variation statistics was used to assess the statistical significance of differences in research results. The obtained data were processed using statistical criteria. Data are presented as the average ± the standard deviation in biological triplicates with a standard error of less than 5%. Analysis of data was performed using one-way ANOVA (n ≥ 3). *p*-values were determined by the two-tailed paired *t*-test at the 5% level of probability.

## 5. Conclusions

Our study aimed at a detailed analysis of the effect of a natural polyphenol with high biological and pharmacological activity. DHQ demonstrates how *Y. lipolytica* yeast can be used as a model organism to detect and elucidate the mechanisms of action of chemical compounds that can slow down aging and delay the onset of age-related diseases in evolutionarily remote eukaryotic organisms. Thus, it is possible to conduct extensive screening of various geroprotective compounds, including those found in natural sources and extracted from certain plant objects. In addition, a number of studies have shown the inhibitory effect of polyphenols on biofilms formed by pathogenic fungi *C. albicans* [82,83].

Taking into account some systematic similarities of *Y. lipolytica* with the pathogenic strain of *C. albicans*, namely, the ability to dimorphic transition and the formation of mycelial and pseudomycelial forms, this model experimental object can be considered as the promising test system for screening potential antimycotic drugs of natural origin, including those based on natural phenolic compounds.

## Figures and Tables

**Figure 1 ijms-25-12574-f001:**
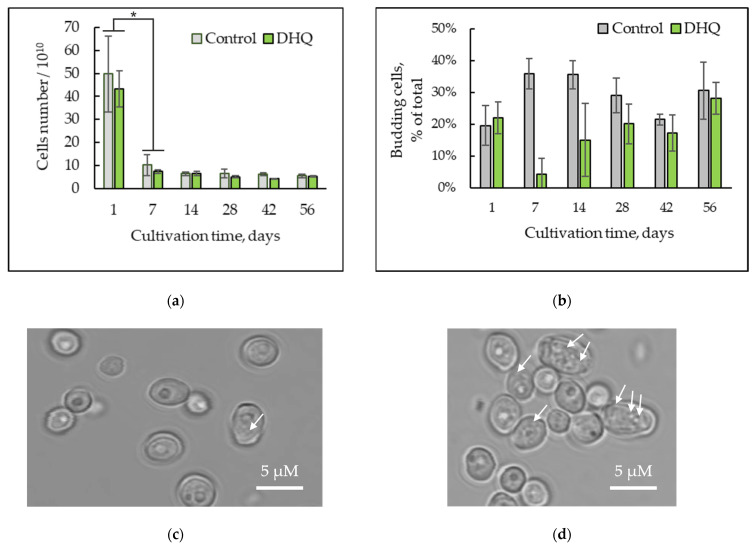
Dynamics of the total number of the cells (**a**) and the share of budding cells (**b**) upon the prolonged cultivation upon addition of DHQ. (**c**,**d**)—Micro images of the cells upon long-lasting cultivation. The photos were taken with an AxioCam MRc camera (magnification 100×). (**c**)—24 h of cultivation; (**d**)—8 weeks of cultivation. The white arrows show vacuoles. *—Statistically significant difference between samples, *p* ≤ 0.05. (**b**) Does not show statistically significant difference. Full statistical analysis is presented in the Appendix A section.

**Figure 2 ijms-25-12574-f002:**
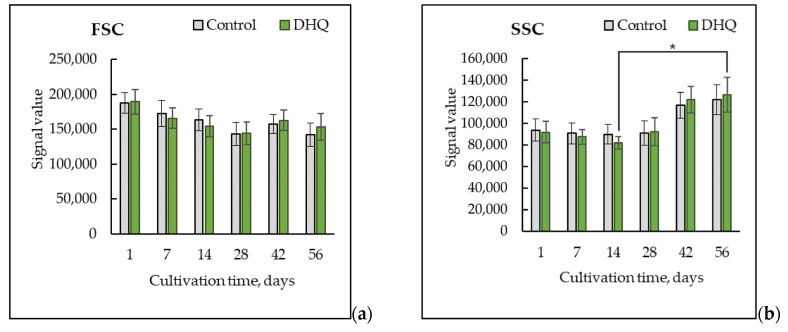
The medians of forward (FSC-A) (**a**) and side (SSC-A) (**b**) scatter of 10 × 10^3^ cells for the control and experimental samples. (**c**–**e**)—Histogram superimposition of forward and side scatter signal upon long-lasting cultivation; (**c**)—24 h of growth; (**d**)—2 weeks of cultivation; (**e**)—8 weeks of cultivation. *—Statistically significant difference between samples, *p* ≤ 0.05. (**a**) Does not show statistically significant difference. Full statistical analysis is presented in the Appendix A section.

**Figure 3 ijms-25-12574-f003:**
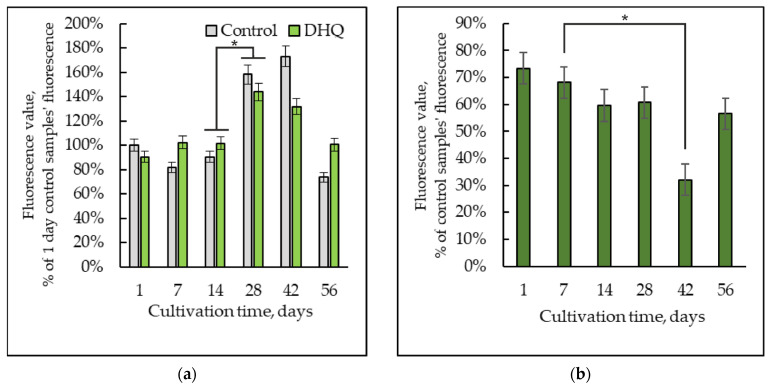
Dynamics of the ROS generation in the control and experimental samples upon treatment with H_2_DCFDA (**a**) and upon the AAPH influence of (**b**), presented as the fluorescence ratio of the samples with DHQ to the control samples of the same age. *—Statistically significant difference between samples, *p* ≤ 0.05. Full statistical analysis is presented in the Appendix A section.

**Figure 4 ijms-25-12574-f004:**
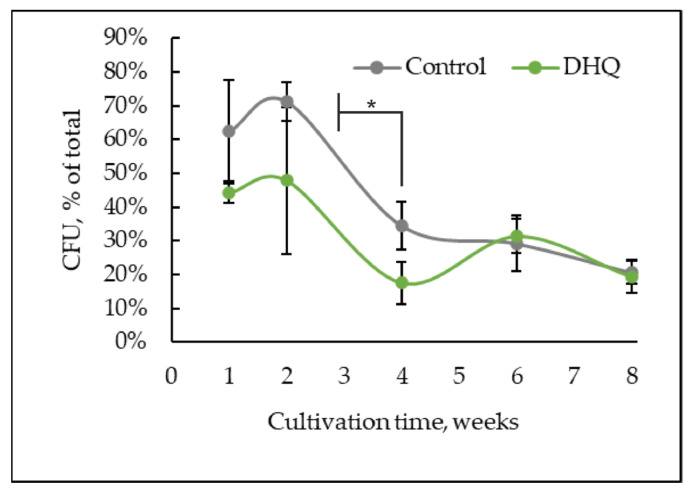
Dynamics of the CFU number in the control and experimental samples upon long-lasting cultivation. *—Statistically significant difference between samples, *p* ≤ 0.05. Full statistical analysis is presented in the Appendix A section.

**Figure 5 ijms-25-12574-f005:**
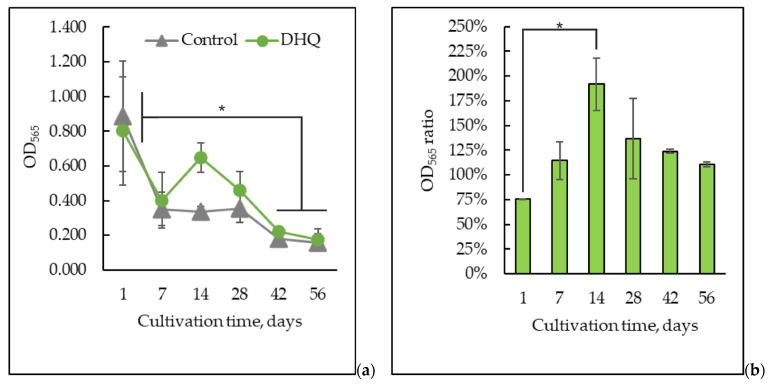
The metabolic activity (MTT test) of the control and experimental samples upon long-lasting cultivation. (**a**) A_590_; (**b**) A_590_ of the experimental samples to the A_590_ of the control samples of the same age. *—Statistically significant difference between samples, *p* ≤ 0.05. Full statistical analysis is presented in the Appendix A section.

**Figure 6 ijms-25-12574-f006:**
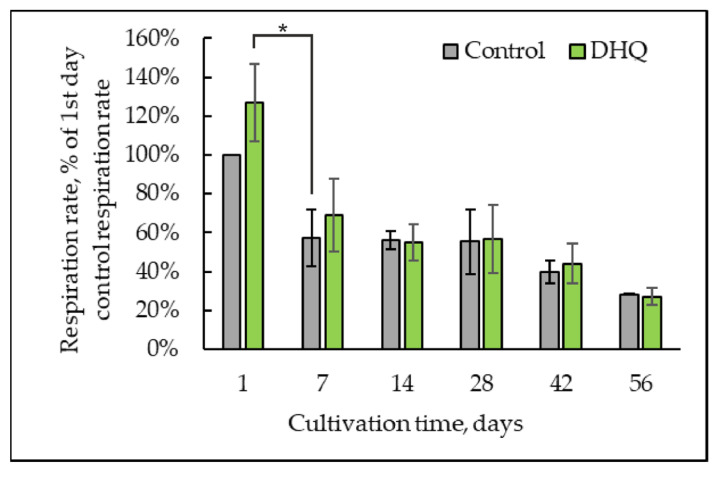
Dynamics of the respiratory rate in the control and experimental samples upon long-lasting cultivation. *—Statistically significant difference between samples, *p* ≤ 0.05. Full statistical analysis is presented in the Appendix A section.

**Figure 7 ijms-25-12574-f007:**
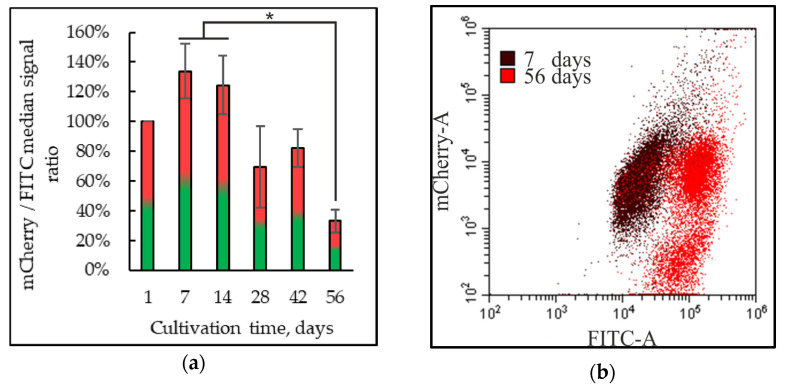
The ratio of the medians of the red fluorescence signal dye to the green one using JC-1 (**a**), superimposition of population fluorescence histograms cultured on the seventh and fifty-sixth days (**b**), the ratio of the mCherry signal to the FITC signal regarding 1 day control + DHQ (**c**), fluorescence microscopy of control (**d**), and experimental (**e**) samples on the seventh day in *Y. lipolytica*. Cells were incubated with 0.5 µM JC-1 for 20 min. The incubation medium contained 0.01 M phosphate-buffered saline (PBS), 1% glycerol, pH 7.4. The areas of high mitochondrial polarization are indicated by bright-red fluorescence due to the concentrated dye. To examine the JC-1-stained preparations, filters 02 and 15 (Zeiss) were used (magnification 100×). Photos were taken using an AxioCam MRc camera. *—Statistically significant difference between samples, *p* ≤ 0.05. Full statistical analysis is presented in the Appendix A section.

**Figure 8 ijms-25-12574-f008:**
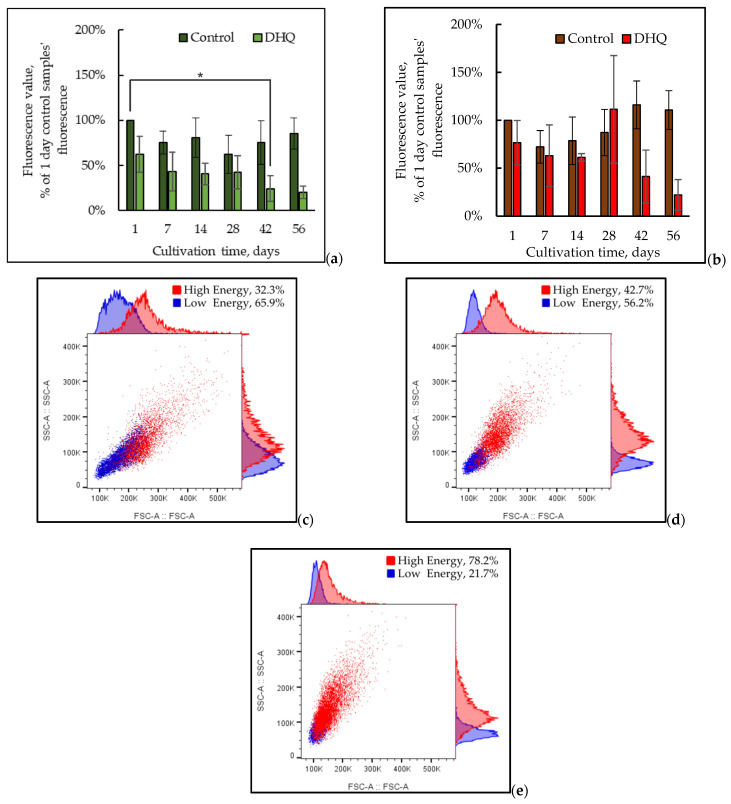
The median of the mitochondrial volume (number of free thiol groups, MitoTracker Green) (**a**) and mitochondrial potential (MitoTracker Red) (**b**) upon long-lasting cultivation. (**c**–**e**) Superimposition of forward and side scatter histograms of the populations cultured for 24 h, 4 weeks, and 8 weeks. (**f**–**h**) Microimages of the cells upon long-lasting cultivation. The photos were taken with an AxioCam MRc camera (magnification 100×); 24 h of cultivation (**f**); 4 weeks of cultivation (**g**); and 8 weeks of cultivation (**h**). *—Statistically significant difference between samples, *p* ≤ 0.05. (**b**) Does not show statistically significant difference.

**Figure 9 ijms-25-12574-f009:**
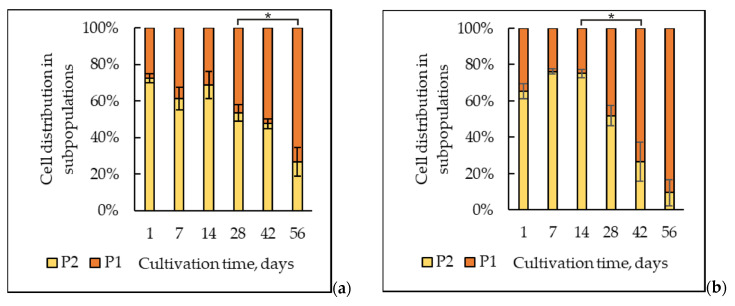
Distribution of the cells into subpopulations with low (P2) and high (P1) mitochondrial activity in the control (**a**) and experimental (**b**) samples. *—Statistically significant difference between samples. Full statistical analysis is presented in the Appendix A section.

**Figure 10 ijms-25-12574-f010:**
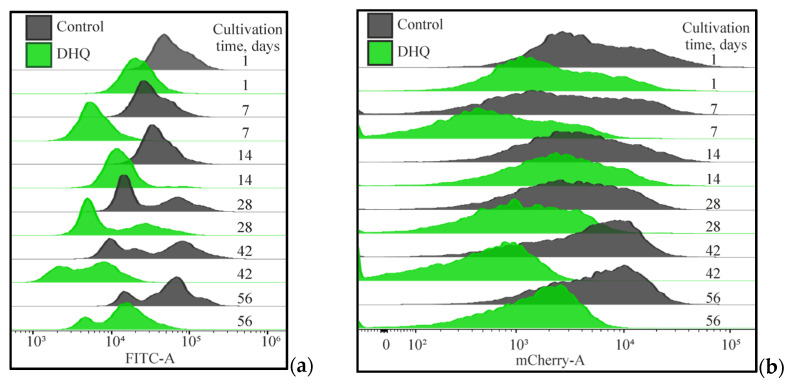
Superimposition of the mitochondrial volume (number of free thiol groups, MitoTracker Green) (**a**) and mitochondrial potential (MitoTracker Red) (**b**) in the control and experimental samples. Full statistical analysis is presented in the Appendix A section.

**Table 1 ijms-25-12574-t001:** Inhibition of respiration by KCN in the control and experimental samples, % to the initial respiratory rate.

Cultivation Time, Days	Inhibition by KCN, %	
	Control	+DHQ
1	40 ± 7 *	44 ± 6 *
7	<10% **	28 ± 8
14	13 ± 9 **	25 ± 11
28	27 ± 11	28 ± 10
42	31 ± 4	13 ± 8 **
56	<10% **	24 ± 14

Means of * and ** are significantly different, *p* ≤ 0.05. Full statistical analysis is presented in the Appendix A section.

## Data Availability

The original contributions presented in the study are included in the article. Further inquiries can be directed to the corresponding author.

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
