# Peer review of "Effect of Dihydroquercetin During Long-Last Growth of Yarrowia lipolytica Yeast: Anti-Aging Potential and Hormetic Properties"

_ijms, 2024, doi:10.3390/ijms252312574_

Round 1
Reviewer 1 Report
Comments and Suggestions for Authors
In the attachment

Author Response
The paper entitled „Effect of dihydroquercetin during long-last growth of Yarrowia lipolytica yeast: the anti-aging potential and hormetic properties” prepared by Maxim S. Pusev et al. contains 22 pages including a title page, abstract, the main text publication divided into chapters, a references, and 10 figures. The authors used a variety of techniques and research methods. Although the results require better presentation, description, and statistical analysis, they lead to interesting conclusions. I recommend this manuscript for publication in „International Journal of Molecular Sciences” after the following revisions.
Answer: We thank the esteemed reviewer for his careful consideration of our study, his time and comments.
Major comments:
- What do the Authors mean by ‘regenerative stress’ and why it is important in the context of dihydroquercetin action?
Answer: We thank the reviewer for identifying the inaccuracy. We meant excessive reductive stress. A detailed description of this phenomenon is given in the study [Reductive Stress in Inflammation-Associated Diseases and the Pro-Oxidant Effect of Antioxidant Agents. Pérez-Torres I, Guarner-Lans V, Rubio-Ruiz ME. Int J Mol Sci. 2017 Oct 5;18(10):2098. doi: 10.3390/ijms18102098]. Reductive stress is an analogue of oxidative stress and can occur under conditions that shift the oxidation-reduction balance of important biological redox pairs, such as NAD+/NADH, NADP+/NADPH, and GSH/GSSG, to a more reduced state. Overexpression of antioxidant enzyme systems results in an excess of reducing equivalents, which can decrease the amount of oxygen radicals, leading cells to reductive stress. Feedback regulation is established, where chronic reductive stress induces oxidative stress, which in turn further stimulates reductive stress. Excess reducing equivalents can regulate cellular signaling pathways, alter transcriptional activity, cause changes in disulfide bond formation in proteins, reduce mitochondrial function, decrease cellular metabolism, and, thus, contribute to the development of some diseases in which NF-B, an oxidative-sensitive transcription factor, is involved. In studying the effect of dihydroquercetin, we consider its antioxidant properties that contribute to the decline of the level of reactive oxygen species in yeast cells, especially in the presence of the exogenous prooxidant 2,2'-azobis (2-amidinopropane) dihydrochloride (AAPH) (Figure 3). Therefore, dihydroquercetin may be involved in processes that shift the oxidation-reduction balance of important biological oxidation-reduction couples, as we can observe in the example of changes in the respiratory activity of yeast mitochondria and the functioning of the alternative oxidase (Figure 6, Table 1).
- Why there is such a drastic decrease in the number of cells after 7 days of culture? As shown, it is not related to a change in the number of budding cells (Fig. 1 a and b). Why do changes in the number of cells not occur gradually over time but suddenly?
Answer: We can assume that 7 days of cultivation may be the point at which the restructuring of metabolic pathways of yeast cells occurs. At this point, substrate depletion is observed, and the cell population undergoes significant changes in the ratio of subpopulations of young and aging cells. It cannot be ruled out that autophagic processes may occur at this stage. Thus, in our previous study [Deryabina, Y.I.; Kokoreva, A.S.; Klein, O.I.; Gessler, N.N.; Isakova, E.P. The Lipid Profile of the Endomyces magnusii Yeast upon the Assimilation of the Substrates of Different Types and upon Calorie Restriction. J. Fungi 2022, 8, 1233. https://doi.org/10.3390/jof8111233] using a specific LysoTracker probe, it was shown that after limiting the calorie content of the culture medium, autophagic processes were observed already at the one-week stage of Endomyces magnusii yeast growth, and after 4 weeks of growth, cells containing autophagosomes accounted for about 50%. It is known that yeast culture demonstrates a decrease in the activity of anabolic processes as a mechanism for survival and energy conservation in response to starvation. However, at the same time, lytic activity in lysosomes enhances to increase degradation reactions and promote catabolic processes of cellular supply [Laidlaw, K.M.E.; Bisinski, D.D.; Shashkova, S.; Paine, K.M.; Veillon, V.A.; Leake, M.C.; MacDonald, C. A glucose-starvation response governs endocytic trafficking and eisosomal retention of surface cargoes in budding yeast. J. Cell Sci. 2021, 134, 257733.]. It is possible that similar processes may take place in the Yarrowia lipolytica culture. Most likely, the decrease in the number of cells does not occur suddenly, but gradually, over 7 days, and we already see the final result of this process. This assumption is confirmed by the fact that the number of cells then remains practically constant up to 8 weeks of growth.
- Why the H2DCFDA probe was used to determine the level of reactive oxygen species? What ROS present in yeast cells during long-term cultivation were determined in this method?
Answer: We have been used the H2DCFDA probe for a long time in our practice of assessing the level of ROS in yeast cells [Isakova EP, Matushkina IN, Popova TN, Dergacheva DI, Gessler NN, Klein OI, Semenikhina AV, Deryabina YI, La Porta N, Saris NL. Metabolic Remodeling during Long-Lasting Cultivation of the Endomyces magnusii Yeast on Oxidative and Fermentative Substrates.Microorganisms. 2020 Jan 9;8(1):91. doi: 10.3390/microorganisms8010091.Deryabina, Y.I.; Kokoreva, A.S.; Klein, O.I.; Gessler, N.N.; Isakova, E.P. The Lipid Profile of the Endomyces magnusii Yeast upon the Assimilation of the Substrates of Different Types and upon Calorie Restriction. J. Fungi 2022, 8, 1233. https://doi.org/10.3390/jof8111233; Rakhmanova, T.I.; Gessler, N.N.; Isakova, E.P.; Klein, O.I.; Deryabina, Y.I.; Popova, T.N. The Key Enzymes of Carbon Metabolism and the Glutathione Antioxidant System Protect Yarrowia lipolytica Yeast Against pH-Induced Stress. J. Fungi 2024, 10, 747. https://doi.org/10.3390/jof10110747] According to several studies, the oxidation of H2DCFDA to 2’-7’dichlorofluorescein (DCF) is widely used for the general detection of ROS, including hydroxyl radicals (•OH) and nitrogen dioxide (•NO2) [LeBel CP, Ischiropoulos H, Bondy SC. Evaluation of the probe 2',7'-dichlorofluorescin as an indicator of reactive oxygen species formation and oxidative stress. Chem Res Toxicol. 1992 Mar-Apr;5(2):227-31. doi: 10.1021/tx00026a012; Reiniers MJ, van Golen RF, Bonnet S, Broekgaarden M, van Gulik TM, Egmond MR, Heger M.Preparation and Practical Applications of 2',7'-Dichlorodihydrofluorescein in Redox Assays. Anal Chem. 2017 Apr 4;89(7):3853-3857. doi: 10.1021/acs.analchem.7b00043.]. Mechanistically, H2DCFDA is taken up by cells where a cellular esterase cleaves off the acetyl groups to form DCFH. Oxidation of DCFH by ROS converts the molecule to DCF, which emits green fluorescence at an excitation wavelength of 485 nm and an emission wavelength of 530 nm. Compared with flow cytometry and other alternative methods, this method using a fluorescence microscope and a plate reader has the advantages of obtaining clearly visible fluorescence images, ease of implementation, high reproducibility in living systems, and cost effectiveness [Szychowski KA, Rybczyńska-Tkaczyk K, Leja ML, Wójtowicz AK, Gmiński J. Tetrabromobisphenol A (TBBPA)-stimulated reactive oxygen species (ROS) production in cell-free model using the 2',7'-dichlorodihydrofluorescein diacetate (H2DCFDA) assay-limitations of method. Environ Sci Pollut Res Int. 2016 Jun;23(12):12246-52. doi: 10.1007/s11356-016-6450-6; Kim H, Xue X. Detection of Total Reactive Oxygen Species in Adherent Cells by 2',7'-Dichlorodihydrofluorescein Diacetate Staining. J Vis Exp. 2020 Jun 23;(160):10.3791/60682. doi: 10.3791/60682; Liu T, Xiao B, Xiang F, Tan J, Chen Z, Zhang X, Wu C, Mao Z, Luo G, Chen X, Deng J. Ultrasmall copper-based nanoparticles for reactive oxygen species scavenging and alleviation of inflammation related diseases. Nat Commun. 2020 Jun 3;11(1):2788. doi: 10.1038/s41467-020-16544-7; Lee S, Ohn J, Kang BM, Hwang ST, Kwon O. Activation of mitochondrial aldehyde dehydrogenase 2 promotes hair growth in human hair follicles. J Adv Res. 2024 Oct;64:237-247. doi: 10.1016/j.jare.2023.11.014.]. In our experiments, we measured total cellular ROS levels using this method, including superoxide anion radical, hydroxyl radical, and hydrogen peroxide.
- Statistical analysis should be performed for all results, not for selected ones.
Answer: We thank the reviewer for an important comment and we have provided a full statistical analysis of our results in the text and in the Supplementary section.
- All results included in the figures should be described in detail. For example, the description of the results shown in Figure 1 concerns only the control conditions. There is no detailed description of the effect of DHQ.
Answer: We thank the reviewer for this note. We have supplemented the section with the necessary information (lines 216-219): “There was no significant effect of DHQ on the total number of cells in the population and the frequency of cell budding, with the exception of the 7-day and 14-day growth stages, where the introduction of DHQ caused a multiple decrease in the number of budding cells by 8- and 2.4-fold, respectively (Figure 1b).”
- The quality of the figures should be improved, for example Figure 10 is completely illegible.
Answer: We have improved the figures, increasing their quality.
Minor comments:
- correct unit record: line 20, line 125
Answer: We have corrected unit record: line 20, line 129
- correct abbreviation notation: line 51 (HIF-1α)
Answer: We have corrected abbreviation notation: line 51 (HIF-1α)
- correct spelling of the name: line 136
Answer: We have corrected spelling of the name: line 151
- correct record of culture density: line 143, line 190, line 191, line 193 etc.
Answer: We have corrected record of culture density
- unfortunate terms: ‘…various pathologies, including aging…’, ‘…great similarity to the mammalian cells’, ‘perhaps, the study…’ etc.
Answer: We have corrected imprecise phrases.
Reviewer 2 Report
Comments and Suggestions for Authors
The manuscript titled „Effect of dihydroquercetin during long-last growth of Yarrowia lipolytica yeast: the anti-aging potential and hormetic properties“ describes a study that aims to demonstrate the feasibility of using yeast as a model organism through which the mechanisms of action of compounds that could slow aging can be elucidated. The work is interesting, current and structured very well. The results are illustrated in great detail and discussed comprehensively. I only have some comments about the layout. Of what origin is the Dihydroquercetin used - synthetic or isolated? On line 136: Should be dihydro-2',7'-dichlorofluorescein instead of dihydro-20,70-dichlorofluorescein. On line 272: Figure 4 should be instead of Figure 5. The sentence on lines 260-262 is not clear whether Figure 4 or 5 refers. Histograms are not visible well. Would the authors consider submitting them as supplementary materials?
Author Response
The manuscript titled „Effect of dihydroquercetin during long-last growth of Yarrowia lipolytica yeast: the anti-aging potential and hormetic properties“ describes a study that aims to demonstrate the feasibility of using yeast as a model organism through which the mechanisms of action of compounds that could slow aging can be elucidated. The work is interesting, current and structured very well. The results are illustrated in great detail and discussed comprehensively.
Answer: We thank the esteemed reviewer for his careful consideration of our work and his high assessment of it.
I only have some comments about the layout.
Of what origin is the Dihydroquercetin used - synthetic or isolated?
Answer: We used the commercial preparation Dihydroquercetin, provided by TransMIT GmbH of the PlantMetaChem Group, Giessen, Germany. We have included this information in the Methods section (lines 138-139)
On line 136: Should be dihydro-2',7'-dichlorofluorescein instead of dihydro-20,70-dichlorofluorescein.
Answer: We have corrected
On line 272: Figure 4 should be instead of Figure 5.
Answer: We thank the reviewer for this comment. The reference to the figure is correct, however, we have corrected the figure number for 4 instead of 5. (Figure 4. Dynamics of the CFU number in the control and experimental samples upon long-lasting cultivation.)
The sentence on lines 260-262 is not clear whether Figure 4 or 5 refers.
Answer: We thank the reviewer for the comment. This is Figure 4. (Dynamics of the CFU number in the control and experimental samples upon long-lasting cultivation.)
Histograms are not visible well. Would the authors consider submitting them as supplementary materials?
Answer: We have tried to improve the quality of the figures and their readability.
Reviewer 3 Report
Comments and Suggestions for Authors
In this study, Pusev et al. investigated the anti-aging potential and hormetic properties of dihydroquercetin (DHQ) on the yeast Yarrowia lipolytica, highlighting its ability to reduce reactive oxygen species (ROS) and influence cell physiology over prolonged cultivation. The study was logically structured, the methodology was clear, and the results were well articulated. However, some points need further clarification. Here are some comments on this study:
1. Line 70 “Saccharomyces cerevisiae” needs to be italicized.
2. Since this study was focused on dihydroquercetin (DHQ), it is recommended that the authors could provide a brief description of the current status of DHQ research in the introduction section.
3. It is suggested that authors emphasize the significance or contribution of the study to the field of research at the end of the introduction.
4. Formatting issues, such as line 115 “ml”, and line 190 “50±16)”.
5. It is recommended that statistically significant differences could be denoted in Figures 1 b and 3b.
6. Figure 2 b-e could be arranged more neatly.
7. It is recommended that the authors provide the p-value of the statistical analysis thourgh the result section, such as line 298 “culture by 40% com- 298 pared to that in the control samples” and Table 1.
Author Response
In this study, Pusev et al. investigated the anti-aging potential and hormetic properties of dihydroquercetin (DHQ) on the yeast Yarrowia lipolytica, highlighting its ability to reduce reactive oxygen species (ROS) and influence cell physiology over prolonged cultivation. The study was logically structured, the methodology was clear, and the results were well articulated.
Answer: We thank the esteemed reviewer for his careful consideration of our work and his high assessment of it.
However, some points need further clarification. Here are some comments on this study:
- Line 70 “Saccharomyces cerevisiae” needs to be italicized.
Answer: We have corrected the Italic font.
- Since this study was focused on dihydroquercetin (DHQ), it is recommended that the authors could provide a brief description of the current status of DHQ research in the introduction section.
- It is suggested that authors emphasize the significance or contribution of the study to the field of research at the end of the introduction.
Answer: We added a brief background on dihydroquercetin to the introduction and highlighted the contribution of our research in this field at the end of the introduction.
- Formatting issues, such as line 115 “ml”, and line 190 “50±16)”.
Answer: We have corrected the text according to this note.
- It is recommended that statistically significant differences could be denoted in Figures 1 b and 3b.
Answer: We have added the necessary information. We have provided a full statistical analysis of our results in the text and in the Supplementary section.
- Figure 2 b-e could be arranged more neatly.
Answer: We tried to arrange the figures 2 b-e more neatly.
- It is recommended that the authors provide the p-value of the statistical analysis thourgh the result section, such as line 298 “culture by 40% com- 298 pared to that in the control samples” and Table 1.
Answer: We have added the necessary information in the text and table.
Round 2
Reviewer 1 Report
Comments and Suggestions for Authors
Once the corrections have been made, the paper can be published in IJMS.